# Potential of Plant Exosome Vesicles from Grapefruit (*Citrus × paradisi*) and Tomato (*Solanum lycopersicum*) Juices as Functional Ingredients and Targeted Drug Delivery Vehicles

**DOI:** 10.3390/antiox12040943

**Published:** 2023-04-17

**Authors:** Alina Kilasoniya, Luiza Garaeva, Tatiana Shtam, Anastasiia Spitsyna, Elena Putevich, Bryan Moreno-Chamba, Julio Salazar-Bermeo, Elena Komarova, Anastasia Malek, Manuel Valero, Domingo Saura

**Affiliations:** 1Cátedra UCAM-DORSIA, Universidad Católica de Murcia, Campus de Los Jerónimos, 30107 Murcia, Spain; 2Petersburg Nuclear Physics Institute Named by B.P. Konstantinov of National Research Centre «Kurchatov Institute», Orlova Roscha 1, 188300 Gatchina, Russia; Garaeva_LAA@pnpi.nrcki.ru (L.G.); Shtam_TA@pnpi.nrcki.ru (T.S.); spitsyna_as@pnpi.nrcki.ru (A.S.); Putevich_ed@pnpi.nrcki.ru (E.P.); 3Institute of Cytology of Russian Academy of Sciences, Tikhoretsky Ave. 4, 194064 St. Petersburg, Russia; elpouta@yahoo.com; 4Instituto de Investigación, Desarrollo e Innovación en Biotecnología Sanitaria de Elche (IDiBE), Universidad Miguel Hernández de Elche, 03202 Alicante, Spain; bryan.morenoc@umh.es (B.M.-C.); m.valero@umh.es (M.V.); 5Subcellular Technology Laboratory, Department of Hematology and Chemotherapy and Department of Radionuclide Diagnostics, N.N. Petrov National Medical Research Center of Oncology, 197758 St. Petersburg, Russia; anastasia@malek.com.ru

**Keywords:** plant exosomes, grapefruit exosomes, tomato exosomes, antioxidant activity, fruit juices, drug delivery

## Abstract

Plant-derived extracellular vesicles (PEVs) have gained attention as promising bioactive nutraceutical molecules; their presence in common fruit juices has increased their significance because human interaction is inevitable. The goal of this study was to evaluate the potential of PEVs derived from grapefruit and tomato juices as functional ingredients, antioxidant compounds, and delivery vehicles. PEVs were isolated using differential ultracentrifugation and were found to be similar in size and morphology to mammalian exosomes. The yield of grapefruit exosome-like vesicles (GEVs) was higher than that of tomato exosome-like vesicles (TEVs), despite the latter having larger vesicle sizes. Furthermore, the antioxidant activity of GEVs and TEVs was found to be low in comparison to their juice sources, indicating a limited contribution of PEVs to the juice. GEVs showed a higher efficiency in being loaded with the heat shock protein 70 (HSP70) than TEVs, as well as a higher efficiency than TEV and PEV-free HSP70 in delivering HSP70 to glioma cells. Overall, our results revealed that GEVs present a higher potential as functional ingredients present in juice and that they exert the potential to deliver functional molecules to human cells. Although PEVs showed low antioxidant activity, their role in oxidative response in cells should be further addressed.

## 1. Introduction

Plant-derived extracellular vesicles (PEVs) are small, membrane-bound vesicles released by plant cells. They resemble the exosomes found in mammalian cells in terms of size and morphology, playing a crucial role in intercellular communication [1,2]. PEVs are typically composed of lipids and proteins and are nanoscale, membrane-enclosed particles [3]. They are formed by the budding of endosomal membranes and are subsequently released into the extracellular space [4,5]. PEVs have been related to a range of physiological processes in plants, including stress response, hormone signaling, and pathogen defense [6].

Studies have highlighted the unexpected functions of PEVs in human gastrointestinal tract cells and have suggested that they maintain cellular homeostasis [7,8,9]. Furthermore, PEVs have been found to be capable of being absorbed by the gastrointestinal tract cells, leading to the expression of functional properties such as anti-inflammatory properties, cytoprotective damage, and intestinal stem cell proliferation [10,11,12]. As PEVs are biogenetically plant-derived vesicles, their architecture, components, and molecular processing may reflect the processes taking place in their parental cells, thus the components of PEVs may contain beneficial agents. PEVs can transport chemical cargo with harmless characteristics into the biological environment and perform multiple functions, such as transmitting signals to recipient cells and recognizing antigen-presentation molecules in cell-to-cell communication [1,2].

Isolated PEVs from plant matrices, such as fruits, leaves, seeds, and roots, have varying physical structures and tissue types. Therefore, there has been a growing interest in fruit juices for advanced strategies for producing PEVs from plant sources, especially in the nutraceutical, cosmeceutical, and therapeutic fields. In this context, fruit juices have been proposed as PEV sources, including oranges, apples, grapes, and ginger, among others. The presence of PEVs in fruit juices may also contribute to the biological properties that juices offer to human health after their consumption [4,6,13], such as nutrients, vitamins, minerals, and bioactive compounds.

The importance of PEVs in fruit juices could enhance their nutritional value and provide additional protection for bioactive compounds to exert their functional properties, such as anti-inflammatory, antiproliferative, and antioxidant properties, contributing to the health benefits associated with consuming fruit juices. Additionally, due to their inherent role in intracellular trafficking, native PEVs are efficiently taken up by recipient cells to which they transfer their lipids, mRNAs, and protein cargo [7,10,14]. PEVs have also been proposed as a novel drug delivery system due to their intrinsic resistance to the acidic gastric environment. PEVs can be loaded with exogenous molecules, such as drugs or other health-promoting substances, or modified for engineered targeting, especially in hepatic cells [2,4,5], making them promising vehicles for ectopic cargo delivery. 

Grapefruits and tomatoes are two widely consumed fruits, meaning that they are available for the big-scale extraction of PEVs. Additionally, both grapefruits and tomatoes are known to be rich in bioactive compounds such as polyphenols, carotenoids, and polysaccharides, which provide health-promoting properties. Additionally, PEVs extracted from grapefruits and tomatoes can be used as a novel drug delivery system due to their intrinsic resistance to the acidic gastric environment. In this study, we investigated the antioxidant and drug delivery properties of PEVs obtained from grapefruit and tomato juices in order to reveal their potential applications in functional foods as well as their ability to target specific cells.

## 2. Materials and Methods

### 2.1. Reagents

A Phosphate buffered saline (10× PBS) solution and a Wheat Germ Agglutinin Alexa Fluor^TM^ 647 (AF647) Conjugate Protein Labeling Kit were purchased from Thermo Fisher (Waltham, MA, USA); while the Recombinant Human HSP70 Protein (Active) was purchased from Abcam (Cambridge, UK) (ab78434). Absolute ethanol, 6-hydroxy-2,5,7,8-tetramethylchroman-2-carboxylic acid (Trolox), 2,2′azino-bis(3-ethylbenzothiazoline-6-sulfonic acid) (ABTS^•+^) and 2,2-diphenyl-1-picrylhydrazy (DPPH) were obtained from Merck (Barcelona, Spain). Milli-Q water was obtained from a purified water system Q-Gard^®^ 1 from Merck Millipore (Darmstadt, Germany) with a resistivity of 18.0 MΩ × cm.

Cell culture media, such as Dulbecco’s Modified Eagle Medium with Nutrient Mixture F12 (DMEM/F12), fetal bovine serum (FBS), gentamycin, trypsin-EDTA, and trypan blue stain were purchased from Biolot (Saint Petersburg, Russia). The glioma cell line (Gl-Tr) was purchased from the Laboratory of Cell Biology (National Research Center, Kurchatov Institute, PNPI, Gatchina, Russia). 

### 2.2. Isolation and Purification of PEVs from Tomato and Grapefruit Juices

Fresh grapefruits and tomatoes were used as PEV sources. These fruits were purchased from a local market in Gatchina, Russia. The juices were extracted using a household juicer (Moulinex Y36-Vitafruit, Alençon, France), then each juice was filtered once using a PEV isolation technique, which was performed according to protocols in other studies [7,15,16]. 

Briefly, the filtered juices of grapefruits and tomatoes were sequentially centrifuged using an Avanti J30-I centrifuge (JA-10 rotor, Beckman Coulter, Brea, CA, USA) at 1500× *g* for 30 min, 3500× *g* for 20 min, 10,000× *g* for 60 min, 16,000× *g* for 60 min, and 10,000× *g* overnight to remove large particles and cellular debris. The supernatant was subjected to ultracentrifugation using a Beckman Optima L-90K ultracentrifuge (Ti 45 rotor, Beckman Coulter, Brea, CA, USA) at 150,000× *g* for 2 h. Then, the supernatant was removed, and the pellet was carefully resuspended in 2 mL of 1× PBS by gentle swaying overnight. The volume was adjusted to 10 mL with 1× PBS and ultracentrifuged at 150,000× *g* for 2 h (Ti 70 rotor, Beckman Coulter, Brea, CA, USA). The resulting pellet was resuspended with 1 mL of 1× PBS for at least 1 h at 4 °C. Final samples of grapefruit exosome-like vesicles (GEVs) and tomato exosome-like vesicles (TEVs) were aliquoted, snap frozen in liquid nitrogen, and stored at −80°C until analysis. 

### 2.3. Nanoparticle Tracking Analysis (NTA)

The sizes and concentrations of GEVs and TEVs in suspensions were determined by NTA using the NanoSight^®^ LM10 (Malvern Instruments, Malvern, UK) with a UV laser (45 mW at 405 nm) using a C11440-5B camera (Hamamatsu photonics KK, Shizuoka, Japan). Recording and data analysis were performed using the NTA software 2.3.

The NTA of GEVs and TEVs was performed by diluting the samples in Milli-Q water between 1000 and 100,000-fold at 25 °C (Camera level: 16, Low threshold: 0, High threshold: 2.015) with a minimum expected size of 30 nm. The following parameters were evaluated during the analysis of records monitored for 30 s: average hydrodynamic diameter, mode of distribution, and concentration of vesicles in the suspension. 

### 2.4. Dynamic Light Scattering (DLS) Analysis

The distribution of GEVs and TEVs in size and their Z-potential were evaluated by DLS analysis. For this, an Avalanche Photodiode Detector (APD) laser correlation spectrometer (Brookhaven Mod 90 Plus) as well as two laser correlation spectrophotometers, Photocor Compact-Z and LKS-3 (OOO Fotokor, Moscow, Russia), were used. Measurements were carried out at 25 °C. Each sample was 100-fold diluted in Milli-Q water, and the particle size distribution was plotted according to the results of three measurements, as well as their Z-potential.

### 2.5. Morphology Analysis by Atomic Force Microscopy (AFM) and Scanning Electronic Microscopy (SEM)

The morphology analysis of GEVs and TEVs was carried out by AFM [17]. Briefly, samples of each PEV suspension in PBS were 50-fold diluted with Milli-Q water, and 5 µL aliquots (about 10^7^ particles) were deposited onto freshly mica nanochips (TipsNano, SPM, Tallinn, Estonia). After drying completely at room temperature, the mica surface was flooded with Milli-Q water to dissolve the salt. The remaining water was removed by drying the nanochips for a 2-h incubation at 37 °C (or 24 h at room temperature). The sample topography measurements were performed in semi-contact mode using the atomic force microscope “NT-MDT-Smena B” with an NSG03 probe (NT-MDT, Moscow, Russia). 

The visualization of GEVs and TEVs was also conducted by SEM. Briefly, samples in PBS were diluted 10-fold with Milli-Q water, and 100 μL aliquots were deposited onto coverslips. Then, samples were fixed with 2.5% glutaraldehyde, washed three times with 0.05% PBS, and dehydrated progressively with 30, 50, 70, 80, 90, and 100% ethanol. Samples were subjected directly to a FESEM system (Sigma 300 VP FESEM, Carl Zeiss, Jena, Germany) for morphology visualization without coating.

### 2.6. Antioxidant Activity of PEVs

To determine the antioxidant power contribution of PEVs in grapefruit and tomato juices, their antioxidant activity was evaluated by ABTS and DPPH methods [18]. The antioxidant activity was measured in a pure PEV solution (100% of concentration determined by NTA), 50% PEVs in Milli-Q water, grapefruit, and tomato juices, as well as juices enriched with 2%, 50%, and 100% concentrations of PEVs by NTA. Juices and PEV-enriched juice samples were ten-fold diluted in Milli-Q water. In both methods, 20 µL of each sample was mixed with 180 µL of ABTS or DPPH solutions for 6 and 15 min, respectively, at room temperature in darkness. The reduction in absorbance was measured at 734 nm for ABTS and 515 nm for DPPH. The antioxidant activity was calculated according to a six-point standard curve of Trolox (R^2^ = 0.996 for ABTS and R^2^ = 0.994 for DPPPH) and expressed as mg of Trolox equivalent per mL of sample (mg TE/mL). 

### 2.7. Loading of PEVs with HSP70 Protein and Loading Efficiency 

A combination of passive and active cargo loading was used [7]. A recombinant human HSP70 protein, labeled with the AF647 Kit, at a final concentration of 0.1 mg/mL was mixed with the suspension of GEVs and TEVs at a final concentration of ~2 × 10^12^ vesicles/mL and incubated overnight at 4 °C. Then, the mixture was sonicated at a frequency of 35 kHz for 15 min by the Bendelin SONOREX SUPER ultrasonic bath (Bandelin Electronic GmbH & Co. KG, Berlin, Germany) at room temperature and incubated for an additional 90 min at 4 °C. To remove the excess of free proteins, the vesicles were centrifugated ten times at 8 °C, 13,000× *g* for 10 min using a 100-kDA filter (Amicon^®^ Pro Purification System Ultra-0.5 Device, Millipore, Burlington, MA, USA). The centrifugation was performed ten times each with 500 µL of 1× PBS. The first and tenth eluates in the filtration procedure (F1 and F10) were used as controls to ensure the loading of PEVs with HSP70. The obtained suspensions of the protein-loaded GEVs and TEVs were established by NTA. The loading efficiency of PEVs with labeled proteins, as well as the efficiency of washing the vesicles from free proteins (F1 and F10), were analyzed by measuring the AF647 fluorescence with a spectrofluorometer (Hitachi F-7000, Tokyo, Japan) at 651/667 nm of emission/excitation.

### 2.8. Cell Viability Assay

To determine the cytotoxicity of TEVs, GEVs, and loaded vesicles with HSP70, Gl-Tr cells were seeded (10^4^ cells per well) in 96-well plates, incubated for 24 h in complete medium, then the medium was replaced with a medium containing 10^6^ particles (GEVs, TEVs, and loaded vesicles) per well. After 48 h of incubation, the proliferation of the cells was tested by the AlamarBlue^®^ Assay (Thermo Fisher, Eugene, OR, USA). The AlamarBlue Cell Viability Reagent was added to the plate according to the manufacturer’s protocol and incubated for two hours. The fluorescence was detected using an EnSpire Multimode Plate Reader (PerkinElmer, Waltham, MA, USA). All experiments were carried out in quintuplicate.

Cell proliferation in real time during cell incubation with PEVs was assessed in the xCELLigence Real-Time Cellular Analysis (RTCA) system (Agilent, Santa Clara, CA, USA). Briefly, 5 × 10^3^ Gl-Tr cells were seeded into wells of a 16X-E plate, and cell adherence and proliferation were monitored using the RTCA system. After 18 h, the culture medium was replaced with a medium containing TEVs, GEVs, and loaded vesicles with HSP70. The cell index (impedance) was assessed every 15 min. The recording was carried out for 72 h. The results were analyzed using the software of the xCELLigence RTCA DP instrument, Software 1.3 (Agilent, Santa Clara, CA, USA). All samples within each experiment were in duplicates. 

### 2.9. In Vitro PEV-Mediated Delivery of HSP70 Protein into Human Cells

Glioma cells (Gl-Tr) were used to study the delivery of exogenous HSP70 protein by PEVs. Briefly, Gl-Tr cells were maintained in the DMEM/F12 culture media supplemented with 10% FBS and 0.1 mg/mL of gentamycin and incubated at 37 °C under a 5% CO_2_ atmosphere. The culture medium was changed every other day.

For protein delivery assays, cells were stained with trypan blue and counted with a LUNA-II^TM^ automatized cell counter (Logos Biosystems, Anyang, Republic of Korea). Then, 10^5^ cells per well were seeded into 24-well plates and incubated for 24 h at 37 °C under 5% CO_2_. In order to deliver the exogenous proteins, purified samples of protein-loaded PEVs as well as 2% of free protein (0.6 µg) were added to new culture media and incubated with recipient cells. The untreated cells were also included as the control to validate the detection of AF647 labeled HSP70 in cells. After 4 h of incubation, cells were trypsinized and analyzed by flow cytometry (CytoFlex, Beckman Coulter, Brea, CA, USA) through the APC channel. No loss of cell viability was detected at the tested concentrations of PEVs and HSP70, according to the MTT assay. 

### 2.10. Statistical Analysis

Statistical analysis was performed using GraphPad Prism (Dotmatics, San Diego, CA, USA), Prism 8.0.2. software. All experiments were carried out in triplicate, and the experimental data were expressed as the mean ± standard deviation unless it is stated otherwise. The unpaired *t*-test or one-way ANOVA followed by multiple comparisons by Dunnett’s or Tukey’s post hoc tests were used where applicable. Statistical significative differences were considered when *p* < 0.05. 

## 3. Results

### 3.1. Characterization of GEVs and TEVs

To assess the concentration and size of tomato and grapefruit vesicles, the methods of nanoparticle tracking analysis (NTA) and dynamic light scattering (DLS) were used, which are typical for studies of vesicles of both animal and plant origin [5,19,20,21]. The concentration of vesicles isolated from 1.5 L of grapefruit and tomato juices through sequential ultracentrifugation was analyzed using NTA (Figure 1A). The yield of PEVs per 100 mL of grapefruit juice was comparable to that of tomato juice (*p* > 0.05), with concentrations of 2 × 10^12^ and 6 × 10^11^ vesicles, respectively. The hydrodynamic size and size mode were also analyzed using NTA (Figure 2B). In both cases, TEVs showed a larger hydrodynamic size (140.00 ± 13.00 nm) and size mode (112.00 ± 12.00 nm) compared to GEVs (*p* < 0.05).

A DLS analysis was also performed to size the GEVs and TEVs. The data from Photocor Compact-Z (Figure 1C) showed two repeated peaks for GEVs: 31.64 ± 5.00 and 139.27 ± 14.11 nm, with 47% and 35% contribution by mass, respectively, while three peaks were observed for TEVs: 14.65 ± 2.83, 54.71 ± 9.54, and 4.23 × 10^4^ ± 5.95 × 10^2^ nm, with 25%, 47%, and 23% contribution by mass, respectively. The DLS analysis by Brookhaven Mod 90 Plus (Figure 1D) showed one major peak for GEVs close to 50 nm and two peaks for the TEVs near 60 and 700 nm. The overall size distribution recorded (Figure 1E) indicated that the TEVs had a larger size range from 140 to 170 nm, as determined by NTA and DLS, while GEVs had a lower size distribution between 86 to 125 nm, as observed by NTA and DLS (*p* < 0.05). The size distribution was also analyzed using AFM, with small but similar sizes determined for both samples (*p* > 0.05). Regarding the Z-potential (Figure 1F), both samples displayed negative values; however, TEVs had a more negative value than GEVs, as determined by Photocor Compact-Z (*p* < 0.01), while a similar Z-potential was observed for both samples by Brookhaven Mod 90 Plus (*p* > 0.05).

### 3.2. Morphological Characterization of GEVs and TEVs

To morphologically characterize the isolated vesicles, GEVs and TEVs were subjected to visualization by AFM and SEM (Figure 2). The surface topology of GEVs (Figure 2A) and TEVs (Figure 2B) were estimated by AFM. Both samples showed individual vesicles of spherical or oval shapes corresponding to the vesicular topology, with an accumulative diameter between 10–20 nm for GEVs and 10–50 nm for TEVs, and vesicular heights between 3–16 and 2–6 nm, respectively. SEM micrographs of GEVs (Figure 2C) and TEVs (Figure 2D) confirmed a round or oval shape of vesicles in both samples with similar diameters as AFM data. 

### 3.3. Contributions of GEVs and TEVs to the Antioxidant Activity in Juices

PEVs of tomato and grapefruit showed to be poor sources of antioxidant activity (Figure 3). For instance, TEVs and GEVs showed less than 0.05 mg TE/mL of antioxidant activity in both DPPH and ABTS assays, while the juice of both fruits showed a stronger antioxidant activity (*p* < 0.001). The low antioxidant activity of pure PEV samples was similar to that reported in 50% PEVs (*p* > 0.05), which indicated that the NVs do not display a strong antioxidant activity.

When TEVs were added to tomato juice, few variations of the antioxidant activity of mixtures were observed in the DPPH assay (Figure 3A); however, an increase in the antioxidant activity of tomato juice by ABTS was noted as the TEV concentration augmented (*p* < 0.01) (Figure 3B). On the other hand, GEV-enriched grapefruit juice showed a slight decrease in its antioxidant activity, while the GEV concentration of grapefruit increased by both assays (*p* < 0.001) (Figure 3C,D).

### 3.4. GEV- and TEV-Mediated Delivery of Exogenous Proteins into Human Gl-Tr Cells

We investigated the loading efficiency of GEVs and TEVs with exogenous cargoes using a passive/active procedure (Figure 4A). Notably, according to the fluorometric analysis of HSP70-AF647, the first filtrate (F1) contained a significant amount of free HSP70-AF647 (52% of the initial amount, *p* < 0.001), while the last filtrate (F10) was protein-free (0.1% of the initial amount, *p* < 0.001). After washing out free proteins, the proportions of the labeled protein loaded to the PEV samples were about 1.1% of the initial amount added to the PEV suspensions before sonication (*p* < 0.001). 

Finally, GEVs and TEVs loaded with HSP70-AF647 were co-cultured with the recipient Gl-Tr cells. The delivery efficiency of protein to the recipient cells by PEVs was determined using flow cytometry (Figure 4B,C). When cells were treated with TEVs loaded with HSP70-AF647, there was a modest increase in fluorescence accumulation compared to the control (*p* < 0.05); however, a higher fluorescence accumulation was observed in cells treated with GEVs loaded with HSP70-AF647 (*p* < 0.001). The heat shock protein HSP70 has the ability to enter mammalian cells without any delivery system [22]. In our experiments, the incubation of human cells in a culture medium containing HSP70-AF647 also led to the accumulation of a fluorescent signal in the cells. However, the accumulation efficiency in human cells of HSP70 delivered as part of grapefruit, but not tomato vesicles, was observed to be significantly higher compared to free HSP70 protein (Figure 4B). Thus, native GEVs can be an effective system for delivering exogenous proteins to human cells. At the same time, TEVs are much less captured by human cells, which is possibly due to their larger size and more negative Z-potential compared to grapefruit vesicles (Figure 1E,F). 

The recombinant human HSP70 was used in these experiments as a model protein to test the efficiency of the delivery of exogenous proteins to human glioma cells using plant vesicles. The cytotoxic effect of the HSP70, PEVs, as well as HSP70-loaded PEVs was determined (Figure 4D,E). According to the results, neither PEVs, HSP70, nor HSP70-loaded PEVs exerted a cytotoxic effect when compared to untreated cells (control). Moreover, the viability of Gl-Tr cells determined in real-time confirmed that both PEVs and HSP70-loaded PEVs do not interfere with the cell’s viability. 

## 4. Discussion

Natural compounds, such as phytochemicals, have been recognized as potential functional ingredients for human health, displaying several biological properties, including antioxidant, anti-obesity, antimicrobial, and antiproliferative properties, among others [18,23,24,25]. The consumption of fruits, vegetables, and derived products, has been associated with a reduction in the risk of chronic diseases [11]; however, the low efficacy of phytochemicals to exert their functional properties after consumption in plant-rich diets as well as supplements has been mentioned [11]. The study of nanosized vesicles from plant sources has recently attracted the interest of researchers because of the ability of these vesicles to shuttle a variety of molecules from the producing cell to the target cells, playing a role in the physiological cell-to-cell communication [5,12,14]. Moreover, their presence in food products, such as fruit beverages, may facilitate the interaction between the recipient cells and the phytochemicals in their external environment [3,23], enhancing their functional effects. 

Here, we assessed the PEVs from grapefruit and tomato, two well-studied fruits with high bioactive content that are highly consumed worldwide [7,8,26]. Native PEVs from grapefruit and tomato juices were characterized by their size, quantity, and morphology by commonly used NTA, DLS, AFM and SEM in order to validate the obtained results. The NTA and DLS methods are commonly used approaches for assessing the concentrations and particle sizes of both animal and plant origin [5,14,19,20,21]. Both methods are most often used to analyze the size of nanoparticles, but a comparison of these methods leads to some differences in the data obtained in a number of studies on extracellular vesicles (EVs) isolated from human biological fluids or culture media [19,27,28]. This disagreement is possibly related to the principles of determining the size of the nanoparticles. The NTA method is based on the tracking of single particles, while the DLS performs the frequency distribution of the reflected light from the particle volume. The presence of several large particles in the solution can make a huge contribution to the total scattering, which leads to a significant frequency shift over large size ranges. It is also worth noting that an overestimation of the particle size may occur due to the measurement of the hydrated diameter of the EV. At the same time, the disadvantage of the NTA method is that during the measurement, large particles in the sample will be perceived by the device as noise and not be detected [20]. Therefore, both methods should be used to correctly estimate the size of such particles. Transmission electron microscopy (TEM) and atomic force microscopy (AFM) can also provide additional information on the size of vesicles, but they are mainly used to assess morphology [29]. AFM is a more accessible method that confirms the presence of particles of a given size and shape in a sample, but due to the peculiarities of sample preparation, the “drying” of particles gives somewhat distorted data on the size of vesicles. Here, a similar concentration of PEVs was found in both grapefruit and tomato juices, while a similar size and morphology determined in exosomes from mammalian and other plant sources were also observed [7,26]. TEVs showed a higher size mode, size distribution, and zeta potential module than GEVs, according to NTA and DLS results. Also, TEVs showed a more heterogenous size distribution than GEVs, with three vesicle sizes, one of them being microscale. Also, in both cases, a smaller size of PEVs was registered by AFM, probably due to the sample preparation that reduced the size of vesicles for microscopy observation, especially the dehydration procedure that the sample requires to be visualized [10]. The spheric morphologies of GEVs and TEVs agreed with the morphology observed by SEM. In this sense, TEVs showed a higher stability in suspension than GEVs, which showed a tendency to aggregate or precipitate due to their Z-potential, however, this should be confirmed by Cryo-EM, as others have mentioned [7,9]. In this sense, TEVs showed to be a more promising source of PEVs regarding their charge; however, GEVs showed a more homogeneous population of PEVs. 

Regarding their biological potential, it has been mentioned that PEVs may contain phytochemicals like ascorbic acid, folic acid, and phenolics such as flavonoids and anthocyanins [8,12], as well as being conformed by lipids that may play a role in reducing the oxidative stress in cells [10]. Here, the antioxidant activity of pure GEVs and TEVs was determined by ABTS and DPPH assays; however, both samples displayed a low antioxidant in both assays, especially when compared to their sources (grapefruit and tomato juices), confirming their poor antioxidant activity. Although PEVs showed low antioxidant activity, their antiradical power related to specific lipids and membrane proteins may display functional effects once they migrate specific cells due to their nanoscale properties [10]. For instance, the composition of PEVs has been related to anti-inflammatory activities in human epithelial and hepatic cells by inhibiting pro-inflammatory interleukin production, as well as mediating the antioxidant activity by reducing ROS production [9,11,12,13]. 

Even though the compositions of GEVs and TEVs should be further studied, it has been mentioned that phosphatidylethanolamine and phosphatidylcholine conform in GEVs [8], while phosphatidic acid and diacylglycerol pyrophosphate may be present in TEVs [30]. Their presence in GEVs and TEVs is probably related to PEVs’ antioxidant activity, upregulating the antioxidant/detoxifying genes such as HO-1, NAO1, GCLM, and GCLC, as observed in liver cells in mice models [31], increasing their importance as antioxidant promoters. 

Due to their lipid content, PEVs have also shown a potential to study antioxidants in biological systems given their hydrophilic/hydrophobic domains [32], increasing their application such as in low-density lipoproteins oxidation studies as well as the study of other lipid-containing substrates delivered to human cells, such as erythrocytes, hepatocytes, and macrophages, among others. In this context, liposomes, or lipids vesicles, such as PEVs, exert characteristics that protect themselves against oxidation in biological systems, protecting compounds or drugs for delivery as drug carriers [32]. 

We also loaded GEVs and TEVs with protein cargoes using HSP70-AF647. HSP70 is an important protein that plays a role in protecting cells from stress-induced damage, increasing cell survival under stress conditions, as well as increasing the sensitivity of cells to chemotherapy [7,33]. Also, the delivery of proteins such as HSP70 to tumor cells faces challenges such as specific cell-type affinity, cellular uptake, intracellular trafficking, and degradation by proteases [34]. In both samples, the loading efficiency was similarly low, however, they were able to retain more proteins than washing the control (F10) between the two samples. GEVs showed a significantly more efficient uptake of HPS70-AF647 by glioma cells compared to free proteins as well as TEVs, according to flow cytometry assay. Several approaches have been explored to overcome the challenges of protein delivery, including the use of exosomes [35]. 

In this case, the high delivery efficiency of GEVs to glioma cells demonstrates the potential of GEVs as efficient carriers of functional exogenous protein into the human cells in vitro; moreover, the HSP70 delivery by GEVs may optimize the cytoprotective effects of this protein and sensitize the tumor cells for chemotherapies, potentially leading to improved treatment outcomes [34]. Also, these results suggest that GEVs can be isolated from grapefruit juices. By loading proteins such as HSP70 into these exosomes, it may be possible to create functional ingredients with potential therapeutic benefits as prophylaxis; however, this potential must be further studied in depth through in vivo and clinical trials.

These exosomes are particularly interesting as vehicles for the controlled dosage of various drugs directly to macrophages and antibody production [4,32]. Therefore, it would be crucial to characterize these PEVs in terms of their phospholipid composition, as it has been reported to be closely related to the antioxidant capacity and to the functionality of exosomes as carriers, especially for delivering RNA to the immune system dendritic cells. The obtained results suggest that PEVs can be isolated from juices and loaded with exogenous proteins for drug delivery.

This study determined the delivery of HSP70 by PEVs in GL-Tr cells; however, further studies using in vivo models should be carried out to determine PEV bioaccumulation, biocompatibility, and gastrointestinal stability, as well as to elucidate their target cells. Additionally, the lipid and protein composition of PEVs may play a role in their stability to fuse with or be internalized by mammalian cells [36], influencing their delivery properties. It has been mentioned that endocytosis pathways are involved in the uptake of human cell exosomes by mammalian cells [37]. Regarding PEVs, they are able to be internalized into mammalian cells through endocytosis pathways related to micropinocytosis and polymerization microtubules [31]. Therefore, further research is needed to determine the composition and internalization mechanisms of both GEVs and TEVs.

A Spearman correlation was performed to determine a relation among the functional properties of PEVs from grapefruit and tomato juices. The results showed a strong correlation between the Z-potential and the delivery efficiency displayed by PEVs and especially GEVs, showcasing the importance of PEV charge for delivery in recipient cells. The obtained results showed the potential of native PEVs for the delivery of exogenous proteins into mammalian cells. Although PEVs from grapefruit and tomato juices exerted low antioxidant activity, their presence in juices as functional ingredients may play a crucial role in the cellular antioxidant response against ROS, which indicates that their bioactive composition should be addressed, especially their lipid and protein profile. Moreover, further efforts should be performed to increase the PEV yield production from fruit matrices in order to take advantage of their full potential.

## 5. Conclusions

In this study, we aimed to examine the antioxidant and drug-delivery properties of PEVs derived from grapefruit and tomato juices. Our goal was to determine their functional potential on their own as well as with vehicles for the delivery of exogenous proteins. The GEVs and TEVs obtained from grapefruit and tomato juices showed similar sizes and morphologies to exosomes from mammalian cells; however, the yield obtained indicated that further improvements are needed to scale up the production of PEVs from fruit juices. The antioxidant activity found in GEVs and TEVs was low; however, it may contribute to the overall antioxidant power of their sources and play a crucial role in modulating the antioxidant response in human cells, which warrants further investigation. Between the two PEV samples, GEVs showed the highest potential for being loaded with the exogenous protein HSP70 and the highest potential for serving as a drug delivery vehicle to glioma cells, highlighting their potential as functional ingredients. However, further research is necessary to fully characterize the composition of PEVs and to understand their biological potential. 

## Figures and Tables

**Figure 1 antioxidants-12-00943-f001:**
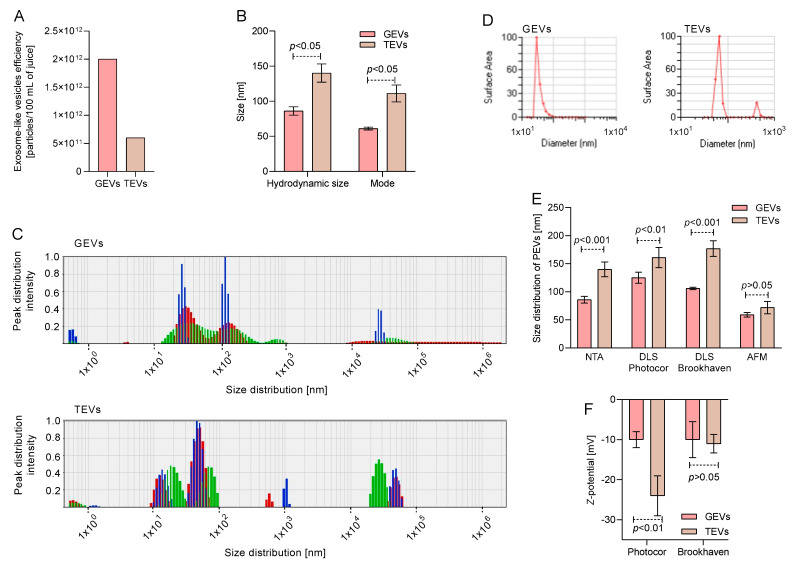
Characterization and particle size of grapefruit exosome vesicles (GEVs) and tomato exosome vesicles (TEVs). Nanoparticle tracking analysis (NTA) of (**A**) concentration as well as (**B**) hydrodynamic size and size mode of GEVs and TEVs. GEVs showed a higher hydrodynamic size and mode distribution than TEVs (*p* < 0.05 with unpaired *t*-student test, n = 3). Representative Dynamic light scattering (DLS) vesicle size distributions of GEVs and TEVs determined by (**C**) Photocor Compact-Z and (**D**) Brookhaven Mod 90 Plus systems. (**E**) Resume of size distribution of GEVs and TEVs by NTA, DLS, and atomic force microscopy (AFM). Overall, TEVs show a higher size distribution than GEVs (*p* < 0.05 with unpaired *t*-student test, n = 3). (**F**) Z-potential of GEVs and TEVs. TEVs show a minor Z-potential than GEVs by Photocor Compact-Z system (*p* < 0.01 with unpaired *t*-student test, n = 3).

**Figure 2 antioxidants-12-00943-f002:**
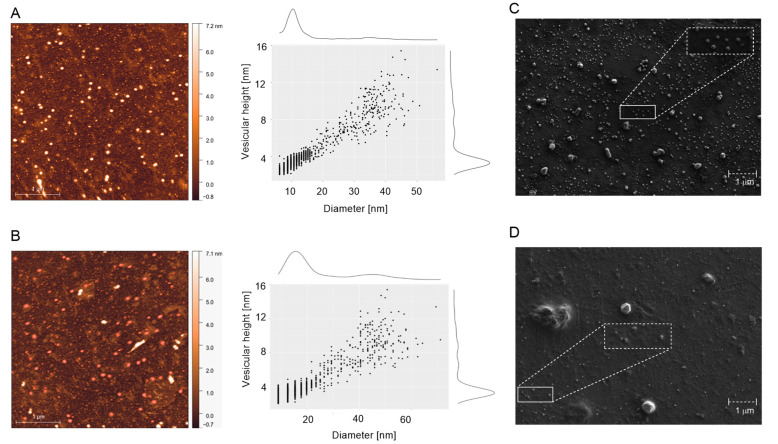
Atomic force microscopy (AFM) and field emission scanning electron microscope (FESEM) micrographs of grapefruit exosome vesicles (GEVs) and tomato exosome vesicles (TEVs). AFM topography scan (left) and cross section (right) of (**A**) GEVs and (**B**) TEVs. On the right of topography scans is the pseudo color ruler indicating the particles’ height (nm). FESEM micrographs of (**C**) GEVs and (**D**) TEVs. A representative group of vesicles has been magnified.

**Figure 3 antioxidants-12-00943-f003:**
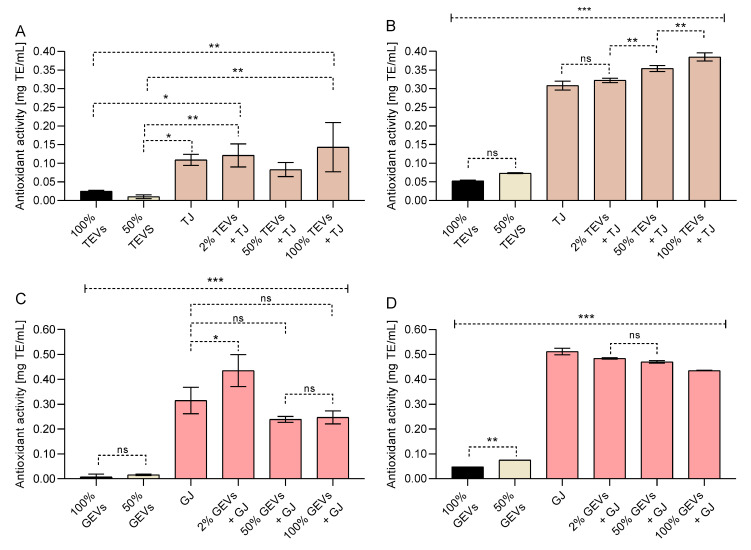
Antioxidant activity by (**A**,**C**) DPPH and (**B**,**D**) ABTS radicals assays of (**A**,**B**) tomato exosome vesicles (TEVs) and (**C**,**D**) grapefruit exosome vesicles (GEVs). Overall, pure GEVs and TEVs showed a lower antioxidant activity than tomato (TJ) and grapefruit juices (GJ) (*** *p* < 0.001, ** *p* < 0.01, * *p* < 0.05, ns *p* > 0.05; One-way ANOVA with Tukey’s post hoc test, n = 3). Results are expressed as mg of Trolox equivalent (TE) per mL of sample ± standard deviation (SD). SD < 0.01 were not plotted.

**Figure 4 antioxidants-12-00943-f004:**
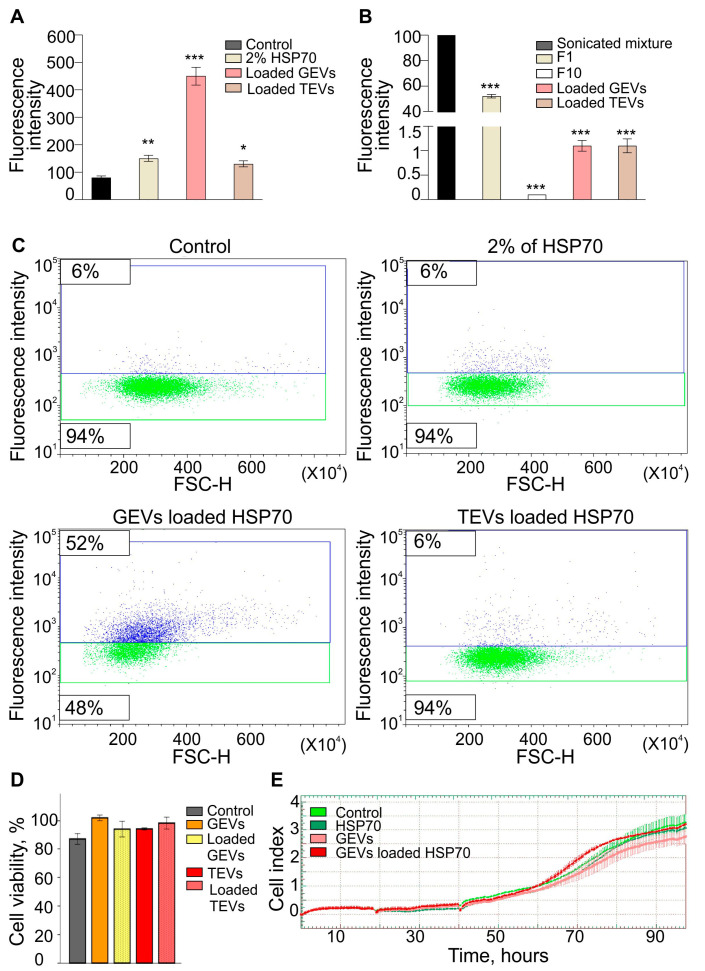
Grapefruit exosome-like vesicles (GEVs) and tomato exosome-like vesicles (TEVs) mediated the delivery efficiency of exogenous HSP-70 proteins to human cells. (**A**) Fluorescence of Alexa Fluor 647 (AF647) labeled-HSP70 in the initial mixture and loaded samples of GEVs and TEVs, as well as washing filtrates (F1 and F10). After loading, excess free proteins were washed out, as shown from F1 to F10. Similar to TEVs, GEVs showed a loading efficiency of 1.10% when compared to the initial amount added to the vesicle suspension (sonicated mixture) (*** *p* < 0.001; One-way ANOVA with Dunnett’s post hoc test, n = 3). (**B**) Fluorescence intensity of the uptake of GEVs and TEVs loaded with HSP70-AF647 by glioma (Gl-Tr) cells. (**C**) Delivery efficiency of protein to recipient cells by GEVs or TEVs analyzed by flow cytometry. Fluorescence signal accumulation into Gl-Tr cells was higher by loaded GEVs than the control (untreated cells) as well as 2% of free protein and TEVs (*** *p* < 0.001, ** *p* < 0.01, * *p* < 0.05; One-way ANOVA with Dunnett’s post hoc test, n = 3). (**D**) Cytotoxicity of grapefruit or tomato vesicles loaded with recombinant HSP70 for Gl-Tr glioma cells. The cytotoxic effects have been studied by the AlamarBlue cell viability assay after 48 h of incubation. No cytotoxic effect was determined when treatments were compared to untreated cells (Control) (*p* > 0.05; One-way ANOVA with Dunnett’s post hoc test). (**E**) Cell index in real-time after treatment of Gl-Tr glioma cells with HSP70, GEVs, or GEVs loaded with HSP70. Control cells were incubated with culture medium only. The average of two replicates is shown for each condition. Error bars represent standard deviations.

## Data Availability

The data are contained within this article.

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
