# Peer review of "Potential of Plant Exosome Vesicles from Grapefruit (Citrus × paradisi) and Tomato (Solanum lycopersicum) Juices as Functional Ingredients and Targeted Drug Delivery Vehicles"

_antioxidants, 2023, doi:10.3390/antiox12040943_

Round 1

Reviewer 1 Report

The manuscript entitled "Potential of plant-exosome vesicles from grapefruit (Citrus x paradisi) and tomato (Solanum lycopersicum) juices as functional ingredients and targeted drug delivery vehicles" presented for review is of great medical and pharmaceutical importance. The manuscript is well written. The purpose of the research was precisely defined. Research goals have been achieved. The obtained research results were discussed in terms of the available scientific literature. However, before publication, the manuscript must be absolutely supplemented with the following aspects:

1) the experimental part must be supplemented with information on the validation of the experimental methods used;

2) the obtained validation parameters of the experimental methods used must be discussed in the 'Discussion' section

Author Response

Dear reviewer,

Thank you for critical reading and valuable suggestions on the improvement of our manuscript. The detailed replies to the comments are below.

Reviewer 1

The manuscript entitled "Potential of plant-exosome vesicles from grapefruit (Citrus x paradisi) and tomato (Solanum lycopersicum) juices as functional ingredients and targeted drug delivery vehicles" presented for review is of great medical and pharmaceutical importance. The manuscript is well written. The purpose of the research was precisely defined. Research goals have been achieved. The obtained research results were discussed in terms of the available scientific literature. However, before publication, the manuscript must be absolutely supplemented with the following aspects:

1) the experimental part must be supplemented with information on the validation of the experimental methods used;

We understand the importance of providing information on the validation of experimental methods uses in our study, and we appreciate the reviewer’s feedback on this matter.

We would like to assure the reviewer that all the experimental methods used in our study were carefully validated to ensure their reliability and accuracy. Our methods have been used extensively in previous studies such as the isolation of PEVs by differential centrifugation, nanoparticle tracking analysis, dynamic light scattering analysis, microscopy visualization, antioxidant activity (DOI: 10.1016/j.foodchem.2013.04.118; DOI: 10.3390/antiox10111668; DOI: 10.1016/j.fct.2010.05.060; DOI: 10.1016/j.foodres.2020.109882; DOI: 10.1097/HEP.0000000000000303; DOI: 10.1016/j.procbio.2018.11.008) of several phytochemicals by different methods, loading of PEVs with HSP70 protein and in vitro delivery into recipient cells (DOI: 10.1038/s41598-021-85833-y; DOI: 10.3390/biomedicines8070216; DOI: 10.21203/rs.3.rs-79427/v1; DOI: 10.1371/journal.pone.0227949; DOI: 10.1038/s41598-021-00734-4; DOI: 10.1371/journal.pone.0242732).

Moreover, to validate our results, we included the analysis of size distribution and Z-potential by three different systems: Brookhaven Mod 90 Plus and Photocor Compact-z, as well as nanoparticle tracking analysis by NanoSight® LM which was used also for the determination of particle concentration in samples. The morphology was also determined by two methods: atomic force microscopy which is a common technique for the evaluation of exosome morphology as well as scanning-electron microscopy which is a common tool for morphology evaluation. The antioxidant activity of samples was determined by both assays such as DPPH and ABTS, which are common techniques used in de evaluation of the antioxidant power of phytochemical samples. Regarding the loading and delivery capacity of PEVs, the assays contained the appropriate controls to determine the quantity of HSP70 encapsulated by PEVs as well as their internalization in recipient cells.

2) the obtained validation parameters of the experimental methods used must be discussed in the 'Discussion' section

In our study native PEVs from grapefruit and tomato juices were characterized by their size, quantity, and morphology by commonly used Nanoparticle tracking analysis (NTA), Dynamic light scattering (DLS), atomic force microscopy (AFM) and Scanning Electronic microscopy (SEM). The NTA and DLS methods are the most commonly used approaches for assessing the concentration and size of particles of both animal and plant origin. Both methods are most often used to analyze the size of nanosized particles, but a comparison of these methods leads to some differences in the data obtained in a number of studies on extracellular vesicles (EVs) obtained from human biological fluids or culture media. This disagreement is possibly related to the principles of determining the size of nanoparticles. The NTA method is based on the tracking of single particles, while the DLS performs the frequency distribution of the reflected light from the particle volume. The presence of several large particles in the solution can make a huge contribution to the total scattering, which leads to a significant frequency shift over large size ranges. It is also worth noting that overestimation of the particle size may occur due to the measurement of the hydrated diameter of the EV. At the same time, the disadvantage of the NTA method is that during the measurement, large particles in the sample will be perceived by the device as noise and not detected. Therefore, both methods should be used to correctly estimate the size of such particles.

Electron microscopy (EM) and atomic force microscopy can also provide additional information on the size of vesicles, but they are mainly used to assess morphology. AFM is a more accessible method that confirms the presence of particles of a given size and shape in a sample, but due to the peculiarities of sample preparation, “drying” of particles gives somewhat distorted data on the size of vesicles. Additional explanations regarding the methods used in the study, as well as additional references, are included in the Discussion section and References sections.

Reviewer 2 Report

General comment: This article aimed to examine the antioxidant and drug delivery properties of PEVs derived from grapefruit and tomato juices. Their goal was to determine their potential applications as functional ingredients and their ability to target specific cells for drug delivery. The main idea is novel and interesting.

Major:

1.     The authors made lots of effort to determine the particle size and other physical characteristics of the isolated PEVS. However, the efficacy of the HSP70-loaded-PEVs against the cell model didn’t explain. Why the loading of protein such as HSP70 into PEVs is a crucial marker in this study?

2.     As mentioned above, the biological activity of the HSP70-loaded-PEVs against the cell model remained unclear. The cell viability of HSP70-loaded-PEVs treated cells should be provided, like MTS assay, MTT assay, CCK-8 assay, or SRB assay.

3.     In Fig. 1E. Why the results of AFM-determined size distribution were different from the other three methods? Please explain or discuss it.

4.     In conclusion, line 384 mentioned the ability to “target specific cells” for drug delivery. The results didn’t indicate this phenomenon. Please explain the soundness of this statement.

Minor:

1.     The term should be carefully checked for accuracy. For instance, in line 153, …PEVs in grape and tomato…, it should be “grapefruit”.

2.     The resolution of figs should be improved. Fig. 4C is hard to read.

Author Response

Dear reviewer,

Thank you for taking the time to review our paper and for the valuable suggestions on the improvement of our manuscript. The detailed replies to the comments are below.

Reviewer 2

General comment: This article aimed to examine the antioxidant and drug delivery properties of PEVs derived from grapefruit and tomato juices. Their goal was to determine their potential applications as functional ingredients and their ability to target specific cells for drug delivery. The main idea is novel and interesting.  

 Major:

1-2     The authors made lots of effort to determine the particle size and other physical characteristics of the isolated PEVS. However, the efficacy of the HSP70-loaded-PEVs against the cell model didn’t explain. Why the loading of protein such as HSP70 into PEVs is a crucial marker in this study? As mentioned above, the biological activity of the HSP70-loaded-PEVs against the cell model remained unclear. The cell viability of HSP70-loaded-PEVs treated cells should be provided, like MTS assay, MTT assay, CCK-8 assay, or SRB assay.

We reply to the first and second comments jointly as we think it might be better to fully understand what we mean.

Hsp70 chaperone is considered in a number of studies as an activator of the antitumor immune response in models in vivo. But this protein does not have a direct negative effect on the growth of tumor cells in culture. In this study, we used human recombinant Hsp70 as a model protein to demonstrate the ability of tomato and grapefruit vesicles to deliver potentially therapeutic exogenous proteins to human glioma cells. Taking into account the fair remark of the reviewer, we analyzed the survival of cells during their incubation in the presence of both types of plant vesicles, free Hsp70, and also this protein in the vesicles from grapefruit and tomato. The cytotoxic effect of both Hsp70 itself, grapefruit or tomato vesicles, as well as Hsp70 protein in the composition of plant vesicles was not detected. These results are added to the Materials and Methods, Results, and Figure 4. Also the importance of Hsp70 use in the study, its relevance in delivery mediated by PEVs into cells and the importance of loading PEVs with Hsp70 have been addressed in Discussion section, lines 391-404.

Regarding to comment 2, the biological activity of HSP70-loaded-PEVs in the cell model has also been addressed in Discussion section, in lines 391-404. Regarding the cell viability assessment, the MTT results showed there were no significant loss of glioma cells viability when exposed to PEVs loaded with HSP70, lines 207-209.

  1. In Fig. 1E. Why the results of AFM-determined size distribution were different from the other three methods? Please explain or discuss it.

Thank you for your comment. Indeed, atomic force microscopy (AFM) can provide information on vesicle size, but is mainly used as an aid or to analyze the shape, morphology, and homogeneity of particles in a sample. AFM clearly confirms the presence of particles of the studied range of sizes and shapes in the sample, but due to the peculiarities of sample preparation, “drying” of particles gives somewhat distorted data on the sizes of vesicles. In addition The differences in size between AFM and the other methods is related to the sample preparation procedure, where the sample requires to be dehydrated in order to be observed under microscopy.. This last issue and other relevant additions have been made to the Discussion section.

  1. In conclusion, line 384 mentioned the ability to “target specific cells” for drug delivery. The results didn’t indicate this phenomenon. Please explain the soundness of this statement.

Thank you very much for the precise remark. We agree, the results do not justify the written claim. Our aim in the study was to determine the possible beneficial potential of plant vesicles on their own, as well as to determine the possibility of their use as containers for the delivery of exogenous molecules. The targeting of such delivery was not at all included in the consideration of this study. Therefore, in the revised manuscript, we have corrected the too loud statements made earlier in the Conclusion.

Minor:

  1. The term should be carefully checked for accuracy. For instance, in line 153, …PEVs in grape and tomato…, it should be “grapefruit”.

Thanks for you for pointing out this. We have corrected typos and errors in the text of the manuscript.

  1. The resolution of figs should be improved. Fig. 4C is hard to read.

In the updated Figure 4, the resolution of panel C has been improved.

Reviewer 3 Report

Dear Editor,

I found the manuscript very interesting since EVs analysis and loading with drug delivery is a key topic in many fields of research. Plant EVs represent a very innovative area and I think the application as vectors for humans could be a challenging issue.

I would like to ask the authors to highlight this vision. I mean, how do they could design a drug vector based on PEV able to work into mammalian tissues? I think that the study in cell lines model is good for a proof of concept but is is not enough in perspective. The membrane composition of PEV will allow the fusion with recipient cell membrane as expected from autologous EVs or it is just a matter of delivering with other mechanisms such phagocytosis/lisosomes-like internalization? Are there any pieces of literature about this?

Author Response

Dear reviewer,

We have proceeded to correct your comments in the manuscript. Thank you for taking the time to review our paper. We appreciate your comments and suggestions to improve the quality of our work. We have addressed your comments and revised our paper accordingly.

Reviewer 3

I found the manuscript very interesting since EVs analysis and loading with drug delivery is a key topic in many fields of research. Plant EVs represent a very innovative area and I think the application as vectors for humans could be a challenging issue.

I would like to ask the authors to highlight this vision. I mean, how do they could design a drug vector based on PEV able to work into mammalian tissues? I think that the study in cell lines model is good for a proof of concept but is not enough in perspective. The membrane composition of PEV will allow the fusion with recipient cell membrane as expected from autologous EVs or it is just a matter of delivering with other mechanisms such phagocytosis/lisosomes-like internalization? Are there any pieces of literature about this?

Thank you for pointing out this. We agree PEVs are a very innovative area of research and as such, they are going to face several challenges for their therapeutic applications. We have proceeded to highlight this in the last part of discussion. The drug vector-based design was addressed in lines 376-382 while PEV composition and internalization mechanism was mentioned in lines 383-393.

Macromolecules enter the intracellular space through the plasma membrane by endocytosis. Macrophages, monocytes, and neutrophils are characterized by phagocytosis, while all other types of animal cells are characterized by pinocytosis, in particular, macropinocytosis for particles larger than 1 µm, clathrin-mediated endocytosis for biomolecules with a size of about 120 nm, caveolin-mediated endocytosis for particles up to 60 nm, as well as clathrin- and caveolin-independent endocytosis [1].

For liposomes, a caveolin-mediated pathway was shown [2], for liposomes derived from lipids of grapefruit vesicles, the blocking of several penetration pathways at once led to a decrease in the efficiency of their uptake by human cells in vitro. This concerns autophagy when using bafilomycin A1, which prevents the maturation of autophagic vacuoles, phagocytosis when cells are treated with cytochalasin D, which is an inhibitor of microfilament formation, and clathrin-mediated endocytosis when cells are exposed to chlorpromazine. While treatment with amiloride, an inhibitor of macropinocytosis, and indomethacin, an inhibitor of the caveolin-mediated pathway characteristic of liposomes, did not affect the absorption of grapefruit-derived liposomes [3].

Partially, these results were confirmed for intact grapefruit vesicles. Thus, it was shown that amiloride, which inhibits the process of macropinocytosis, reduces the uptake of GEVs by 50%, and chlorpromazine, which suppresses clathrin-mediated endocytosis, reduces the uptake of grapefruit vesicles by more than 60% [4].

Thus, at the moment, based on a few studies, clathrin-mediated endocytosis can be assumed for plant-derived vesicles as pathways of penetration into mammalian cells, while the involvement of macropinocytosis and phagocytosis requires further research.

 [1] D. Conner & Sandra L. Schmid. Regulated portals of entry into the cell Sean. Nature. 2003 Mar 6;422(6927):37-44. doi: 10.1038/nature01451.

[2] Pollock, S., Antrobus, R., Newton, L., Kampa, B., Rossa, J., Latham, S., Branza Nichita, N., Dwek, R. A., Zitzmann, N. Uptake and trafficking of liposomes to the endoplasmic reticulum. FASEB J. 24, 1866 –1878 (2010).

[3] Qilong Wang, Xiaoying Zhuang, Jingyao Mu, Zhong-Bin Deng, Hong Jiang, Lifeng Zhang, Xiaoyu Xiang, Baomei Wang, Jun Yan, Donald Miller, Huang-Ge Zhang. Delivery of therapeutic agents by nanoparticles made of grapefruit-derived lipids. Nat. Commun. 4:1867 doi: 10.1038/ncomms2886 (2013).

[4] Baomei Wang, Xiaoying Zhuang, Zhong-Bin Deng, Hong Jiang, Jingyao Mu, Qilong Wang, Xiaoyu Xiang, Haixun Guo, Lifeng Zhang, Gerald Dryden, Jun Yan, Donald Miller, Huang-Ge Zhang. Targeted Drug Delivery to Intestinal Macrophages by Bioactive Nanovesicles Released from Grapefruit. Mol Ther. 2014 Mar; 22(3): 522–534. doi: 10.1038/mt.2013.190

Round 2

Reviewer 1 Report

The authors revised the manuscript in accordance with the comments in my review and responded to these comments.